# Hyperpolarized ^13^C-Pyruvate Metabolism as a Surrogate for Tumor Grade and Poor Outcome in Renal Cell Carcinoma—A Proof of Principle Study

**DOI:** 10.3390/cancers14020335

**Published:** 2022-01-11

**Authors:** Stephan Ursprung, Ramona Woitek, Mary A. McLean, Andrew N. Priest, Mireia Crispin-Ortuzar, Cara R. Brodie, Andrew B. Gill, Marcel Gehrung, Lucian Beer, Antony C. P. Riddick, Johanna Field-Rayner, James T. Grist, Surrin S. Deen, Frank Riemer, Joshua D. Kaggie, Fulvio Zaccagna, Joao A. G. Duarte, Matthew J. Locke, Amy Frary, Tevita F. Aho, James N. Armitage, Ruth Casey, Iosif A. Mendichovszky, Sarah J. Welsh, Tristan Barrett, Martin J. Graves, Tim Eisen, Thomas J. Mitchell, Anne Y. Warren, Kevin M. Brindle, Evis Sala, Grant D. Stewart, Ferdia A. Gallagher

**Affiliations:** 1Cancer Research UK Cambridge Centre, University of Cambridge, Cambridge CB2 0QQ, UK; su263@cam.ac.uk (S.U.); rw585@cam.ac.uk (R.W.); mary.mclean@cruk.cam.ac.uk (M.A.M.); mireia.crispinortuzar@cruk.cam.ac.uk (M.C.-O.); cara.brodie@cruk.cam.ac.uk (C.R.B.); abg28@cam.ac.uk (A.B.G.); marcel.gehrung@cyted.ai (M.G.); lucian.beer@meduniwien.ac.at (L.B.); jlb1004@cam.ac.uk (J.F.-R.); surrin.deen@addenbrookes.nhs.uk (S.S.D.); f.riemer@web.de (F.R.); jk636@cam.ac.uk (J.D.K.); f.zaccagna@gmail.com (F.Z.); jd906@cam.ac.uk (J.A.G.D.); mjl99@cam.ac.uk (M.J.L.); af557@cam.ac.uk (A.F.); iosif.mendichovszky@addenbrookes.nhs.uk (I.A.M.); sjw236@cam.ac.uk (S.J.W.); tb507@cam.ac.uk (T.B.); tgqe2@cam.ac.uk (T.E.); tjm@sanger.ac.uk (T.J.M.); anne.warren@addenbrookes.nhs.uk (A.Y.W.); kmb1001@cam.ac.uk (K.M.B.); es220@cam.ac.uk (E.S.); gds35@cam.ac.uk (G.D.S.); 2Department of Radiology, University of Cambridge, Cambridge CB2 0QQ, UK; anp11@cam.ac.uk (A.N.P.); james.grist@dpag.ox.ac.uk (J.T.G.); 3Department of Radiology, Addenbrooke’s Hospital, Cambridge University Hospitals NHS Foundation Trust, Cambridge CB2 0QQ, UK; mjg40@cam.ac.uk; 4Department of Urology, Addenbrooke’s Hospital, Cambridge University Hospitals NHS Foundation Trust, Cambridge CB2 0QQ, UK; antony.riddick@addenbrookes.nhs.uk (A.C.P.R.); tevita.aho@addenbrookes.nhs.uk (T.F.A.); james.armitage@addenbrookes.nhs.uk (J.N.A.); 5Department of Endocrinology, Addenbrooke’s Hospital, Cambridge University Hospitals NHS Foundation Trust, Cambridge CB2 0QQ, UK; rc674@cam.ac.uk; 6Department of Oncology, Addenbrooke’s Hospital, Cambridge University Hospitals NHS Foundation Trust, Cambridge CB2 0QQ, UK; 7Department of Surgery, University of Cambridge, Cambridge CB2 0QQ, UK; 8Wellcome Sanger Institute, Hinxton CB10 1RQ, UK; 9Department of Pathology, Addenbrooke’s Hospital, Cambridge University Hospitals NHS Foundation Trust, Cambridge CB2 0QQ, UK

**Keywords:** cancer metabolism, hyperpolarized ^13^C magnetic resonance imaging, monocarboxylate transporter, renal cell carcinoma

## Abstract

**Simple Summary:**

We evaluated renal cancer with varying aggressive appearances on histology, using an emerging form of non-invasive metabolic MRI. This imaging technique assesses the uptake and metabolism of a breakdown product of glucose (pyruvate) labelled with hyperpolarized carbon-13. We show that pyruvate metabolism is dependent on the aggressiveness of an individual tumor and we provide a mechanism for this finding from tissue analysis of molecules influencing pyruvate metabolism, suggesting a role for its membrane transporter.

**Abstract:**

Differentiating aggressive clear cell renal cell carcinoma (ccRCC) from indolent lesions is challenging using conventional imaging. This work prospectively compared the metabolic imaging phenotype of renal tumors using carbon-13 MRI following injection of hyperpolarized [1-^13^C]pyruvate (HP-^13^C-MRI) and validated these findings with histopathology. Nine patients with treatment-naïve renal tumors (6 ccRCCs, 1 liposarcoma, 1 pheochromocytoma, 1 oncocytoma) underwent pre-operative HP-^13^C-MRI and conventional proton (^1^H) MRI. Multi-regional tissue samples were collected using patient-specific 3D-printed tumor molds for spatial registration between imaging and molecular analysis. The apparent exchange rate constant (*k*_PL_) between ^13^C-pyruvate and ^13^C-lactate was calculated. Immunohistochemistry for the pyruvate transporter (MCT1) from 44 multi-regional samples, as well as associations between MCT1 expression and outcome in the TCGA-KIRC dataset, were investigated. Increasing *k*_PL_ in ccRCC was correlated with increasing overall tumor grade (ρ = 0.92, *p* = 0.009) and MCT1 expression (r = 0.89, *p* = 0.016), with similar results acquired from the multi-regional analysis. Conventional ^1^H-MRI parameters did not discriminate tumor grades. The correlation between MCT1 and ccRCC grade was confirmed within a TCGA dataset (*p* < 0.001), where MCT1 expression was a predictor of overall and disease-free survival. In conclusion, metabolic imaging using HP-^13^C-MRI differentiates tumor aggressiveness in ccRCC and correlates with the expression of MCT1, a predictor of survival. HP-^13^C-MRI may non-invasively characterize metabolic phenotypes within renal cancer.

## 1. Introduction

Renal cell carcinoma (RCC) is the twelfth commonest solid cancer globally and the sixth in the United Kingdom [1]. Clear cell RCC (ccRCC) is the most frequent histological subtype found in 75–80% of patients with RCC [2]. In the localized disease stage, partial or radical nephrectomy is the treatment of choice [3]. However, with the increasing detection of incidental small renal masses by cross-sectional imaging, the risk of overtreatment is increasing, because neither conventional imaging nor tissue biopsies predict aggressive tumor behavior reliably [4,5]. Although tumor grade is a leading predictor of progression, recurrence, and survival, tumor biopsies may underestimate the grade owing to sampling error [6,7]. An imaging biomarker that could non-invasively grade ccRCC and exclude differential diagnoses would reduce overtreatment and could have significant clinical impact. This imaging biomarker may be derived from novel image analysis methods and machine learning approaches such as those in radiomics: the extraction and evaluation of high-dimensional quantitative data from images have shown promise in the grading of ccRCC [8,9]. Alternatively, new imaging techniques may provide a direct readout of tumor biology and in this study, we explored whether novel methods for probing tumor metabolism could be used to phenotype renal cancer more accurately.

Renal cell cancer is driven by dysregulation of several pathways involving cellular energy metabolism [10]. Dysregulation of hypoxia-inducible factor 1-alpha (HIF-1α) pathways resulting from VHL mutation or promotor hypermethylation is present in 91% of ccRCC leading to profound alterations in glucose metabolism [11]. Increasing tumor grade is associated with increased lactate concentrations and expression of proteins involved in glycolysis and lactic acid fermentation [12,13]. Metabolomic analyses require invasive sampling of vascularized tissue and are subject to sampling error. In contrast, non-invasive imaging enables repeated, non-destructive assessment of comparatively large volumes of tissue. For example, positron emission tomography (PET) using the glucose analog [^18^F]FDG, can indirectly probe these alterations. However, [^18^F]FDG-PET cannot distinguish metabolites, provides no direct information on glycolytic flux, and is limited in RCC by the background urinary signal [14].

Hyperpolarized carbon-13 magnetic resonance imaging (HP-^13^C-MRI) is a new clinical method to quantify tissue metabolism [15]. The technique can image the conversion of HP-[1-^13^C]pyruvate to [1-^13^C]lactate, catalyzed by lactate dehydrogenase (LDH), and is a promising pre-clinical and clinical imaging biomarker of aggressiveness and early treatment response [16]. HP-^13^C-MRI has been used to image the normal human brain and heart, as well as cancers of the prostate, breast, kidney, pancreas, and brain [17,18,19,20,21]. Monocarboxylate transporters (MCTs) are involved in cellular energy metabolism through the transport of pyruvate, lactate, and ketone bodies across the plasma membrane [22]. Findings from human breast and prostate cancer have identified MCT1 expression as a rate-limiting factor for ^13^C-lactate formation in HP-^13^C-MRI [20,23]. In this proof-of-principle study, we demonstrate that HP-^13^C-MRI provides a non-invasive measure of tumor aggressiveness in ccRCC from multiregional biopsies and show that the imaging measures of metabolism correlate with the expression of MCT1 for the first time in participants with RCC. We subsequently validated these changes in gene expression and patient outcome data from The Cancer Genome Atlas Kidney Renal Clear Cell Carcinoma (TCGA-KIRC) database [24].

## 2. Materials and Methods

### 2.1. Recruitment and Ethics

Study participants were recruited at Addenbrooke’s Hospital, Cambridge University Hospitals NHS Foundation Trust, Cambridge UK and provided written informed consent for this ethically approved study protocol: MISSION Substudy in Renal Cancer (Molecular Imaging and Spectroscopy with Stable Isotopes in Oncology and Neurology; Cambridge South Research Ethics Committee, 15/EE/0378) and DIAMOND (Evaluation of biomarkers in urological diseases; 03/018). The inclusion criteria were: age ≥18 years, clinical suspicion of T1b + RCC, candidate for surgery, and Eastern Cooperative Oncology Group (ECOG) performance status ≤1. Key exclusion criteria were metabolic disorders, allergy to gadolinium-based contrast agents, pregnancy, and unsuitability for MRI. Patients may have harbored localized or metastatic disease as long as surgery was clinically indicated, and they have not received any pre-surgical treatment.

### 2.2. Hyperpolarized [1-^13^C]Pyruvate MRI Acquisition

The preparation of the ^13^C-pyruvate has been described previously [20] and are summarized in the Appendix A. Hyperpolarized ^13^C images were acquired on a clinical 3 T MR system (MR750, GE Healthcare, Waukesha, WI, USA) using a clamshell coil for ^13^C-transmission. A ^13^C-tuned 8-channel array coil (Rapid Biomedical, Rimpar, Germany) was centered over the tumor. An Iterative Decomposition with Shifted Echo times and Least Squares Estimation (IDEAL) spiral chemical shift imaging (CSI) sequence [25] was employed to acquire five axial slices of 3 cm thickness with a 5 mm gap. The center of one slice was placed through the center of the tumor and as much of the tumor and contralateral normal kidney as possible were covered (Appendix A). Radiofrequency pulses with a nominal flip angle (α) of 15° were applied to acquire a 34 cm × 34 cm field-of-view (FOV) with a matrix size of 40 × 40 voxels and a temporal resolution of 4 s for 20 time points (repetition time 0.5 s), starting 12 s after the injection of HP-pyruvate. Images had a true resolution of 17 mm × 17 mm and were reconstructed to 128 × 128 voxels. The patients were instructed to hold their breath for approximately the first 20 s and to breathe gently thereafter.

### 2.3. ^13^C-MRI Data Analysis

The IDEAL spiral data were reconstructed in MATLAB. Complex imaging data acquired over the eight channels of the abdominal array coil were combined using a singular value decomposition approach similar to Chen et al. [26]. For metabolite signals, the complex data were summed over all time points prior to coil combination to minimize noise propagation.

Polygonal regions of interest (ROIs) of the tumor and normal kidney were drawn manually on co-registered T1-weighted images and propagated to metabolic maps using OsiriX 10.0 (Pixemo SARL, Bernex, Switzerland). For individual biopsies, cylindrical ROIs with a diameter and height of 3 cm were centered on the biopsy. Voxel-wise intensity values were exported from OsiriX using the JavaScript Object Notation (JSON) format. Estimation of the mean and standard deviation (S.D.) of the noise was performed in an extracorporeal region of the image where spiral artifacts were absent. The signal-to-noise ratio (SNRmetabolite) was calculated from maps of pyruvate, lactate, and pyruvate+lactate (total carbon) within each ROI was defined as follows (see Formula (1)):(1)$${SNR}_{metabolite}=\frac{\mathrm{mean}\left({\mathrm{SI}}_{ROI}\right)-\mathrm{mean}\left({\mathrm{SI}}_{noise}\right)\;}{\sqrt{2}\;S.D.\;({\mathrm{SI}}_{noise})}$$
where mean (SIROI) is the mean signal intensity in the ROI, and the mean and S.D. of SI_*noise*_ are computed over the ROI containing background only. The factor of √2 accounts for the narrowed Rayleigh distribution of magnitude noise, with an approximate adjustment for the use of multiple receivers.

The apparent reaction rate constant for the exchange of the HP-^13^C label between pyruvate and lactate (*k*_PL_) was calculated using a two-site exchange model in a frequency-domain approach and linear least-squares fitting. Back-conversion of lactate-to-pyruvate (*k*_PL_) and spin-lattice relaxation effects (T_1_) were combined as an effective relaxation term T_1_eff [27].

Masks containing only voxels with a summed carbon signal-to-noise ratio (SNR) of ≥5 were created to avoid voxels with a poor fit.

### 2.4. Proton (^1^H) MRI

The multiparametric ^1^H-MRI was acquired immediately after the HP-^13^C-MRI, following repositioning of the patient into a 32-channel cardiac array coil (GE Healthcare, Waukesha, WI, USA) on the same 3 T MRI scanner. The protocol included diffusion weighted, blood-oxygenation level dependent, and dynamic contrast-enhanced MRI in addition to T_1_- and T_2_-weighted sequences. The sequence parameters in Appendix A were used for all except one participant where the presence of tumor thrombus in the intrahepatic inferior vena cava required a focused pre-surgical assessment (Appendix A). The sequences and image post-processing are described in the Appendix A. All imaging data were analyzed blinded to the clinical outcome.

### 2.5. D-Printed Patient Specific Tumor Molds

Tumor molds were created as described previously [28]. In brief: the tumor, adjacent normal appearing kidney, and the perirenal fat including Gerota’s fascia were segmented manually in OsiriX (Pixmeo SARL, Switzerland). The segmentation was performed on the T_1_w LavaFlex sequence because this provided the highest spatial resolution. The segmentation was re-oriented using in-house MATLAB code and the mold generated using in-house code implemented in Python, interfacing with Meshlab and OpenSCAD. The 3D model was sliced in PrusaSlicer and a Prusa i3 MK3S was used to print the mold in polylactic acid (PLA).

### 2.6. Histology and Immunohistochemistry

Following nephrectomy, the specimen was sliced using a patient-specific 3D-printed tumor mold to allow co-registration of tissue samples and imaging [28]. The location of biopsies was recorded on maps rendered to represent the cross-section of the tissue at the location of the tissue slice and then transferred to the corresponding location in the T_1_w LavaFlex reference sequence (Appendix A). Three to nine representative tumor samples (6 mm punches of fresh tissue) were obtained depending on tumor size, as well as samples from the ipsilateral normal appearing renal tissue. These were formalin-fixed and paraffin embedded. A subspecialized pathologist determined tumor grade on hematoxylin and eosin (H&E) stained slides according to the WHO/ISUP scale [6], assigning a grade to each tissue sample. Tissue samples with <75% viable tumor tissue were excluded.

Immunohistochemical (IHC) staining for MCT1 and MCT4 was performed using Leica’s Polymer Refine Detection System (DS9800) in combination with their Bond automated system (Leica Biosystems Newcastle Ltd., Newcastle upon Tyne, UK).

Sections were cut to 4 μm thickness and incubated with Tris EDTA for 20 min for antigen retrieval. Endogenous peroxidase activity was quenched using 3–4% (*v*/*v*) hydrogen peroxide. The primary antibodies were HPA003324 for MCT1 and HPA021451 for MCT4 (both Atlas Antibodies, Bromma, Sweden) both at a dilution of 1:500. The sections were then incubated in Anti-rabbit Poly-HRP-IgG polymer (<25 μg/mL) containing 10% (*v*/*v*) animal serum in Tris-buffered saline /0.09% ProClin 950 (Sigma-Aldrich, Gillingham, UK). The complex was visualized using 66 mM 3,3′-Diaminobenzidine tetrahydrochloride hydrate in a stabilizer solution and ≤0.1% (*v*/*v*) Hydrogen Peroxide. Leica DAB Enhancer was added to enhance staining. Cell nuclei were counterstained with <0.1% hematoxylin.

HALO v2.2.1870.15 (Indica Labs, Albuquerque, NM, USA) and the area quantification v2.1.11 module were used for automated analysis of scanned sections. Areas of weak, moderate, and strong staining were summed and divided by the total tissue area to obtain the percentage of positive tissue for MCT1 and MCT4 expression. Optical densities for weak, moderate, and strong stains used for the automated quantitative analysis of scanned sections were 0.2164, 0.3274 and 0.4938, respectively. The median percentage of positive tissue of all biopsies from one patient was taken to compensate for spatially heterogeneous marker expression within tumors. Similarly, cell density was quantified using HALO by dividing the number of cells counted in each biopsy by the tissue area. The median for all biopsies from one patient was used for patient-level comparisons.

### 2.7. TCGA-KIRC Data

Findings in the prospective dataset were validated using normalized RNAseq by Expectation Maximization (RSEM) data from the TCGA-KIRC dataset of ccRCC (accessed through Broad Institute, https://gdac.broadinstitute.org/, accessed on 24 January 2021). Gene expression data was z-transformed and the Kruskal–Wallis test used to test if the expression was dependent on tumor grade, complemented by pairwise Wilcoxon rank-sum tests with correction for multiple comparisons (Benjamini–Hochberg [29]). The proportion of variance in the expression of genes explained by another gene was measured by linear regression. Associations of gene expression and overall or disease-free survival (OS and DFS) were explored with Kaplan–Meier analysis at a pre-specified cut-off at the 85th percentile. Univariate and multivariate Cox regression models including clinical variables were fit.

### 2.8. Statistical Analysis

Statistical analyses were conducted using R (version 4.0.0, R foundation for statistical computing, Vienna, Austria). Mean and standard deviation were used as summary statistics for (approximatively) normally distributed continuous variables. Median and range were preferred with skewed data. Comparisons between unpaired samples used the Wilcoxon rank-sum test, paired samples the Wilcoxon Signed-Rank test and correlations the Spearman correlation for skewed data. Comparisons between approximately normally distributed data samples used Student’s t-test and Pearson’s correlation. Linear mixed models and two-way ordinal analysis of variance (ANOVA) were used to account for dependencies of biopsy-level data. The dplyr (version 1.0.2), ggplot2 (3.3.3), GGally (2.1.0) survival (3.2–7), survminer (0.4.8) packages were used for data processing, figure design and survival analysis.

## 3. Results

### 3.1. Participants

Nine participants with suspected renal masses were recruited as part of this physiological study (Appendix A). Participant characteristics are reported in Table 1. All patients were imaged with HP-^13^C-MRI and multiparametric ^1^H-MRI shortly before tissue sampling (median of 12 days; interquartile range: 10–21). Diagnoses of the malignant tumors were confirmed on post-nephrectomy histology. Six patients were diagnosed with clear cell renal cell carcinoma (ccRCC); a dedifferentiated renal liposarcoma and a pheochromocytoma were imaged as they were thought to be RCCs based on pre-operative imaging. One participant, recruited for comparison purposes, harbored a benign oncocytoma which was diagnosed with multi-core needle biopsy. Research tissue was available for all malignant tumors except the pheochromocytoma where all samples were used to establish the clinical diagnosis.

### 3.2. Hyperpolarized ^13^C-MRI

Time-summed spectra of the slice containing the largest cross-section of the tumor and overlays of the [1-^13^C]pyruvate and [1-^13^C]lactate signals summed over time on T_1_w images for all nine patients are displayed in Figure 1. Quality control parameters of the ^13^C-pyruvate preparation are summarized in Appendix A. A median signal-to-noise ratio (SNR) for ^13^C-pyruvate of 26.7 (16.8–61.3) was measured in ccRCC (*n* = 6), similar to the contralateral normal kidney with a median SNR of 30.1 (14.8–64.7; *n* = 8). There was no significant difference in the median ^13^C-lactate SNR between ccRCC and the normal kidney: 5.7 (1.9–9.6) and 3.4 (2.0–9.3; *p* = 0.51), respectively. The dedifferentiated liposarcoma, pheochromocytoma, and oncocytoma showed a median pyruvate SNR of 31.9, 34.0, and 12.3, and a lactate SNR of 10.5, 6.0, and 1.0, respectively.

There was no significant difference in the median lactate-to-pyruvate ratio (Lac/Pyr) between ccRCC (0.13; 0.10–0.42) and normal kidney (0.14; 0.12–0.22; *p* = 0.40). The Lac/Pyr was 0.35 for the liposarcoma, 0.17 for the intrarenal pheochromocytoma, and 0.14 for the oncocytoma. The median *k*_PL_ was 0.0065 (0.0024–0.0151) in ccRCC and 0.0043 (0.0028–0.0076) in the normal kidney. The liposarcoma was the most metabolically active tumor (*k*_PL_: 0.0152; pheochromocytoma: 0.0086). The oncocytoma presented with the lowest *k*_PL_ (0.0022). The pyruvate and lactate SNR were not correlated, however the Lac/Pyr and *k*_PL_ correlated strongly (r = 0.88, *p* = 0.03, Figure 2).

An increasing median *k*_PL_, in contrast to Lac/Pyr, correlated with a higher WHO/ISUP tumor grade (ρ = 0.92, *p* = 0.009; Figure 3a; Appendix A). The median *k*_PL_ for individual biopsies (*n* = 44) was strongly patient-dependent (*p* < 0.001). A higher *k*_PL_ in individual biopsies remained significantly predictive of higher tumor grade after accounting for dependence (*p* = 0.03, Figure 3b). Similarly, the Lac/Pyr was significantly increased in higher-grade tumor samples (*p* = 0.005, Appendix A).

Appendix A shows an example of ^13^C- and ^1^H-MRI obtained from one participant. Comparison of ^1^H-MRI parameters with WHO/ISUP tumor grade showed that perfusion fraction, a measure of capillary volume, trended towards a negative association with grade, but failed to reach statistical significance (Appendix A). No other ^1^H-MRI parameter correlated with tumor grade. The Lac/Pyr negatively correlated with the apparent diffusion coefficient D_0_ (*p* = 0.003, r = −0.98; Figure 2b, Appendix A). The perfusion fraction showed a trend for correlation with the median *k*_PL_ (*p =* 0.30, r = −0.58; Figure 2c).

Tumor volume showed no association with WHO/ISUP grade (Appendix A) and ^13^C-MRI.

### 3.3. Immunohistochemistry

MCT1 expression in ccRCC was positively correlated with the *k*_PL_ (r = 0.89, *p* = 0.016; Figure 4), Lac/Pyr (r = 0.94, *p* = 0.005; Appendix A) and grade. The *k*_PL_ showed borderline significance in predicting the variance in tumor MCT1 expression at the biopsy level (*p* = 0.052; Appendix A) and was independent of the cell density in tumor biopsies (*p* = 0.95, Appendix A). MCT4 expression was not correlated with metabolic activity (Appendix A).

The median MCT1 for all samples in each patient showed a trend towards decreased expression in ccRCC compared to normal renal tissue which was not significant (4.0 ± 6.0% vs. 6.2 ± 3.4%, *p* = 0.17, *n* = 6). Median MCT4 showed a higher positive pixel count in ccRCC compared to normal renal tissue (37 ± 10.5% vs. 10.2 ± 6.0%, mean ± S.D.; *p* = 0.009, *n* = 6).

The dedifferentiated liposarcoma showed a lower expression of MCT4 compared to ccRCC (16% vs. 37%) and an MCT1 expression comparable to WHO/ISUP grade 4 ccRCC (5.4% vs. 6.2%).

The concentration of serum LDH measured immediately before the HP-^13^C-MRI was independent of the tumor aggressiveness (Appendix A). Measurements were not available for two participants owing to hemolyzed samples.

### 3.4. TCGA-KIRC RNA Expression

Analysis of RNA expression from 526 patients with ccRCC in the TCGA showed that MCT1 and MCT4 expression was grade dependent (*p* < 0.001 and 0.008 respectively) with overexpression in higher grade tumors (Figure 4 and Appendix A). LDHA expression was independent of tumor grade (*p* = 0.06, Appendix A). While the expression of all three genes was highly correlated (all *p* < 0.001), R^2^_adj_ was low (MCT1/MCT4: 0.03, MCT1/LDHA: 0.16, MCT4/LDHA: 0.22; Appendix A).

150 patients reached the overall survival endpoint while the surviving patients were censored after a median of 49 months (inter quartile range (IQR): 27–73 months). One-hundred and fifty patients reached the overall survival endpoint while the surviving patients were censored after a median of 49 months (inter quartile range (IQR): 27–73 months). A total of 127 patients progressed while the non-progressive patients were followed-up for a median of 45 months (IQR: 20–64 months). Univariate Cox and Kaplan–Meier analysis demonstrated that MCT1 and MCT4 expression was predictive of OS, while MCT1 and LDHA were predictive of progression-free survival (PFS; Figure 4, Appendix A). In a multivariate Cox regression model, the presence of metastasis, age at diagnosis, and MCT1 expression, were independent predictors of OS (Table 2). In contrast, only metastasis, nodal stage and grade were predicted DFS (Appendix A).

## 4. Discussion

This proof of principle study has demonstrated the positive correlation between the apparent kinetic rate constant for the conversion of pyruvate to lactate (*k*_PL_) measured using HP-^13^C-MRI, the WHO/ISUP tumor grade and MCT1 expression in patients with ccRCC.

Previous work in RCC cell lines has shown increased ^13^C-lactate efflux in a more aggressive cell, which was attributed to increased MCT4 expression as the predominate mediator of lactate export and accompanied by a reduced MCT1 expression [30]. In vivo experiments showed an association between increasing HP pyruvate-to-lactate flux and higher LDHA expression [30,31,32]. Proteomic and metabolomic analyses of human RCCs confirm the relationship between lactate and tumor grade [12,13], and demonstrate the wide range of potential molecular drivers of HP-^13^C-lactate signal, raising the question of what may be the predominant factor in human ccRCC in vivo. A recent clinical study of HP-^13^C-pyruvate MRI in six patients with ccRCC (two additional patients had chromophobe RCCs), showed an increasing overall Lac/Pyr was associated with a higher tumor grade [21]. Here we have analyzed 44 multiregional biopsies from six ccRCC patients comparing biopsy grade with the co-registered metabolic imaging using a 3D tumor mold and compared immunohistochemical measures of pyruvate metabolism with imaging to explain the underlying molecular mechanism. The present study also assessed the complementary value of HP-^13^C-MRI and ^1^H-MRI and the value of the absolute quantification of the metabolic activity using *k*_PL_.

Multiparametric ^1^H-MRI was used to characterize the tumors morphologically and physiologically. The perfusion fraction was most strongly associated with tumor grade, but this association was weaker than for *k*_PL_, and did not reach statistical significance. The independence of perfusion fraction and *k*_PL_ suggests that the metabolic changes detected between tumor grades are likely to reflect alterations in cellular metabolism rather than a perfusion effect. Although RCC volume was not related to metabolic activity in this study, larger and more hypoxic tumors have been shown to be more metabolically active in breast cancer [20].

The expression of MCT1 on IHC, the membrane transporter facilitating pyruvate influx [33], correlated with *k*_PL_ and Lac/Pyr. The latter association has also been demonstrated in breast and prostate cancer [20,23]. The overexpression of MCT1 in grade 4 ccRCC in the TCGA-KIRC database strengthens this finding and, as MCT1 also predicts OS, it underlines its potential prognostic significance which is in agreement with existing literature [34,35]. Other biological factors may also control the formation of the labelled lactate, such as the expression of the enzyme LDH, the concentration of cofactor NADH/NAD^+^, as well as endogenous lactate levels [36].

The results from ^1^H-MRI supports the more glycolytic phenotype of aggressive tumors: the decreased perfusion fraction may indicate that aggressive tumors outgrow their blood supply and rely more on glycolysis. However, while the negative correlation of the perfusion fraction with tumor grade has been described previously, there is debate in the literature with other studies finding no association [37,38].

ccRCC and the contralateral normal kidney showed similar levels of HP-^13^C-pyruvate metabolism, which is supported by the similar MCT1 expression in RCC and normal kidney. MCT1 expression has been shown to be an important determinant of pyruvate-to-lactate conversion in preclinical models of pancreatic and breast cancer [39]. In this study, MCT4 was overexpressed in the tumor compared to normal renal tissue, in keeping with existing data [40]. MCT4 exports lactate into the extracellular space, which is central to maintaining a high glycolytic activity and promoting tumor cell and invasion [41,42]. Owing to the high metabolic activity of the normal kidney, HP-^13^C-MRI is unlikely to be used for detection of renal tumors determine tumor aggressiveness, as these tumors will usually have been detected using conventional imaging techniques as part of routine workflow; major clinical unmet needs include improved non-invasive tumor and treatment stratification, particularly if a biopsy cannot be obtained easily. Therefore, imaging tumor metabolism using HP-^13^C-MRI, will be complementary to the structural and functional information provided by imaging modalities such as CT and ^1^H-MRI.

Alterations in tumor metabolism, and cellular energy metabolism in particular, are a hallmark of cancer and have been exploited as biomarkers of treatment response [43,44]. In vivo studies of HP-^13^C-pyruvate MRI in many cancer models, including ccRCC, have shown treatment-induced metabolic changes as early as 24 h after the start of treatment [45]. Recent evidence from a clinical study has shown that early treatment response to neoadjuvant therapy can be assessed with HP-^13^C-pyruvate MRI after 7–11 days, which is associated with pathological complete response in patients with breast cancer [36]. Detecting successful treatment response earlier will play an increasingly important role in ccRCC with the advent of an increasing number of treatment options in advanced disease.

Future clinical research will assess the clinical utility of HP-^13^C-pyruvate MRI for these key questions of tumor grading and treatment response detection in ccRCC. For example, metabolic imaging could be used to prioritize small but metabolically active renal lesions for surgery rather than surveillance.

Although the number of participants in this study is relatively small, the use of multiregional biopsies has enhanced the power to detect significance between the imaging metrics and the histological and immunohistochemical measurements. The multiregional analyzes employing a 3D-printed tumor mold for accurate co-registration of imaging and tissue samples, confirming the association between *k*_PL_, Lac/Pyr, and tumor grade, as well as *k*_PL_ and MCT1 expression. Two of the patients whose pre-surgical diagnosis suggested the presence of RCC were diagnosed with an intrarenal pheochromocytoma and a dedifferentiated liposarcoma, presenting the first opportunity to investigate the metabolism of these tumors with HP-^13^C-MRI. RCC exhibits marked intratumoral heterogeneity [46] and therefore tissue samples and molecular biomarkers may be subject to sampling error. The non-invasive identification of this intratumoral metabolic heterogeneity represents one of the strengths of this emerging technique and could provide important new measures to assist tumor stratification in the future.

## 5. Conclusions

The apparent kinetic rate constant for the conversion of pyruvate-to-lactate derived from HP-^13^C-MRI (*k*_PL_) differentiated overall ccRCC tumor grade in each tumor, as well as the individual tumor grade acquired from multiregional biopsies in each tumor, in this proof of principle study. This *k*_PL_ was correlated with the expression of the MCT1 pyruvate transporter, which is an independent predictor of survival in patients with ccRCC. Therefore, metabolic imaging may have important prognostic implications for stratifying ccRCCs, could be used to identify tumor metabolic heterogeneity, and could be used to monitor therapy response in the future.

## Figures and Tables

**Figure 1 cancers-14-00335-f001:**
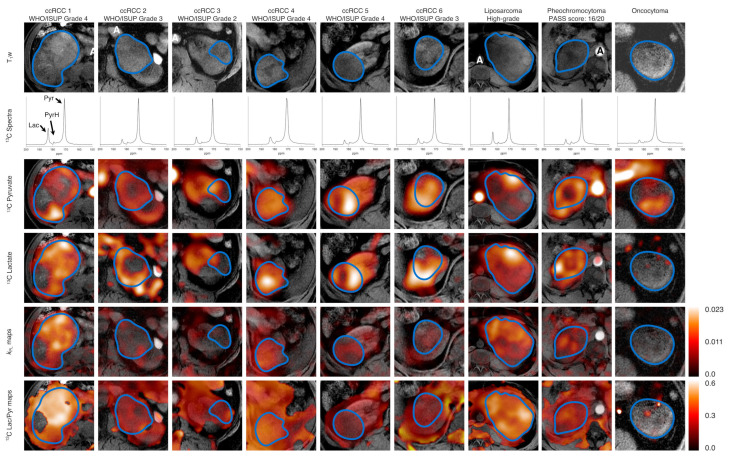
Summary of all tumors included in this study. Axial T_1_w images as a reference. Non-localized ^13^C spectra of the axial slice containing the largest tumor cross-section summed over all timepoints. ^13^C-Pyruvate and ^13^C-lactate signal summed over all time points superimposed on an axial T_1_w image of the largest tumor cross-section. The border of the tumor is outlined in blue. A: Aorta, Lac: Lactate, Pyr: Pyruvate, PyrH: Pyruvate hydrate. A PASS score (Thompson, The American Journal of Surgical Pathology: May 2002) of ≥4 is associated with a potential for clinically aggressive behavior in pheochromocytoma. ^1^H-MRI.

**Figure 2 cancers-14-00335-f002:**
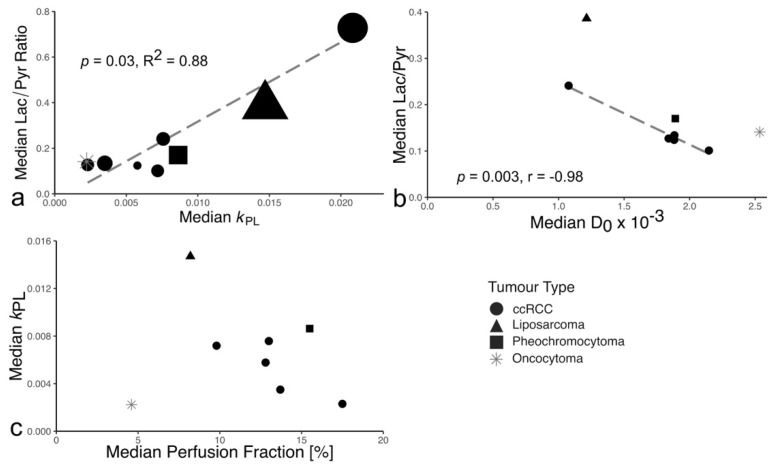
Relationships between hyperpolarized ^13^C-pyruvate MRI parameters as well as between HP-^13^C-MRI and proton MRI parameters. (**a**) The median lactate-to-pyruvate ratio was strongly correlated to the median *k*_PL_. The volume of the tumors, represented by the area of the points on the plot, ranged from 53 to 1350 cm^3^ (53–908 cm^3^ for ccRCC). There was no correlation between ccRCC volume and lactate-to pyruvate ratio and *k*_PL_. (**b**) The median lactate-to-pyruvate ratio was negatively correlated with the median diffusivity. Correlation coefficients calculated for ccRCC only. (**c**) Association of the median perfusion fraction from IVIM-type DWI with the median *k*_PL_ from ^13^C-MRI.

**Figure 3 cancers-14-00335-f003:**
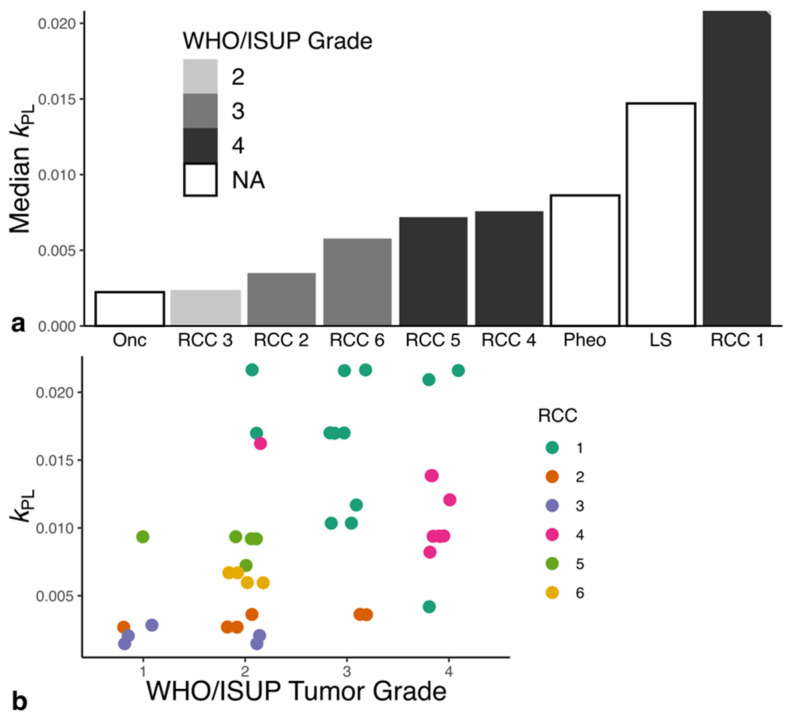
Comparison of the ISUP/WHO Grade with median *k*_PL_. (**a**) Increasing tumor grade was associated with increased metabolic activity in ccRCC. The liposarcoma and pheochromocytoma showed metabolic activity comparable to grade IV ccRCC. The benign renal oncocytoma showed the lowest metabolic activity. (**b**) An increase in *k*_PL_ values was associated with an increasing tumor grade determined for the individual tumor biopsy. LS: Liposarcoma, Onc: Oncocytoma, Pheo: Pheochromocytoma, RCC: clear cell renal cell carcinoma.

**Figure 4 cancers-14-00335-f004:**
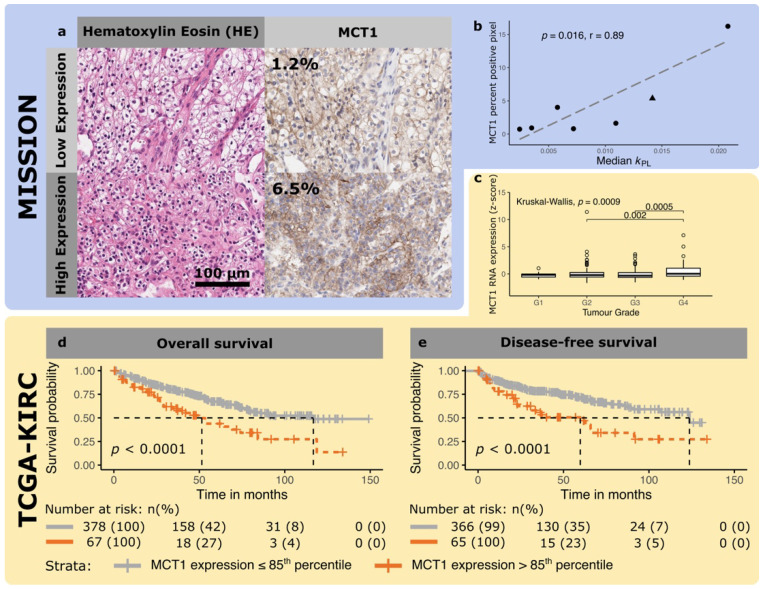
Representative micrographs of a clear cell renal cell carcinoma with high and low expression of MCT1. (**a**) The left panels show a hematoxylin and eosin stain, and the right panels the corresponding immunohistochemical stain for MCT1 on an adjacent tissue slice. (**b**) Correlation between MCT1 IHC expression and *k*_PL_ in the patient dataset imaged as part of this study (MISSION, dots represent ccRCC, the triangle the liposarcomas) (**c**) Box plot comparing z-transformed MCT1 expression as a function of histological tumor grade from a Cancer Genome Atlas dataset (TCGA-KIRC, yellow). (**d**) Kaplan–Meier plots for the association of MCT1 expression with overall and (**e**) progression/recurrence-free survival using an expression cut-off at the 85th percentile in the TCGA-IRC dataset. MCT1 expression was significantly associated with overall and progression/recurrence-free survival.

**Table 1 cancers-14-00335-t001:** Patient characteristics.

Patient Characteristics	Distribution
Patients (male/female)	9 (8/1)
Patient Age (median ± IQR) (years)	59.5 ± 8.7
Histology	6 clear cell renal cell carcinoma 1 Pheochromocytoma 1 Dedifferentiated Liposarcoma 1 Renal Oncocytoma
Tumor Stage at Surgery (RCC only)	2 pT1b pNx cM0 3 pT3a pN0 cM0 1 pT3b pNx cM1
WHO/ISUP Tumor Grade at Surgery (RCC only)	1 Grade 2 2 Grade 3 3 Grade 4
Location of Metastasis	1 Lung
Patient Weight (median ± IQR) (kg)	90.1 ± 13.5
Time between imaging and surgery (median ± IQR) (days)	12 ± 11
Laterality	5 left/4 right
Plasma glucose (median ± IQR) (mmol/l)	5.0 ± 0.3
Patients (male/female)	9 (8/1)

S.D.: standard deviation, IQR: Inter-quartile range.

**Table 2 cancers-14-00335-t002:** Multivariate logistic regression for overall survival in the TCGA-IRC dataset.

Covariates	*p* Value	HR	95% Confidence Interval	
			Lower	Upper
Age	0.002	1.034	1.012	1.058
Female Sex	0.17	0.696	0.416	1.162
LN	0.30	1.579	0.665	3.748
Metastasis	<0.001	3.179	1.834	5.510
Size	0.36	1.191	0.818	1.734
Grade				
Grade 1	0.99	0.001	0	∞
Grade 2	0.15	0.591	0.287	1.218
Grade 3	0.50	0.799	0.414	1.540
MCT1	0.010	1.309	1.065	1.609
MCT4	0.55	1.073	0.850	1.355
LDHA	0.082	0.797	0.617	1.029

HR: Hazard ratio, LDHA: Lactate dehydrogenase A, LN: Lymph node, MCT1/4: Monocarboxylate transporter ¼. Concordance index = 0.783.

## Data Availability

The data presented in this study are available on request from the corresponding author. The data are not publicly available due to ethical restrictions.

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
