# Peer review of "Hyperpolarized 13C-Pyruvate Metabolism as a Surrogate for Tumor Grade and Poor Outcome in Renal Cell Carcinoma—A Proof of Principle Study"

_cancers, 2022, doi:10.3390/cancers14020335_

Round 1

Reviewer 1 Report

Ursprung et al performed 13C-pyruvate imaging and correlative 1H imaging, histological and gene expression analyses on nine patients with renal masses who went on to receive nephrectomies. They performed metanalyses using the TCGA-KIRC dataset to assess MCT1 gene expression level and survival. The overall assertion is that calculation of the tumor Kpl value using HP-13C-pyruvate imaging correlated with tumor aggressiveness and with the expression of MCT1. Although the sample numbers are far too small to make any such statistically significant conclusions, the trend that was observed between higher Kpl value correlating with higher WHO/ISOP grade in the six ccRCC tumors included in this study shows promise that this in vivo imaging modality may have predictive clinical indications for RCC patients in the future, and justifies the expansion of such studies to additional RCC patients.

  • Use of a 3D-printed mold for spatial accuracy of sample acquisition from the tissue specimens is interesting.
  • The first paragraph of the discussion section is overly wordy and quite lengthy. Needs to be edited for brevity.
  • Discussion section needs to (further) highlight the small sample size utilized here and that this is a proof of principle study that lacks the number of samples needed to make any final conclusions.
  • In lines 275-277, the authors state that, “Comparison of 1H-MRI parameters with WHO/ISUP tumor grade showed that perfusion fraction, a measure of capillary volume, was negatively associated with grade (Supplementary Figure A3d). No other 1H-MRI parameter correlated with tumor grade.” However, in the abstract of this manuscript, it is stated that, “Conventional 1H-MRI parameters did not discriminate tumor grades.” This discrepancy has to be corrected since the data suggest that tumor grade was in fact correlated with perfusion fraction, which is a conventional 1H-MRI measure.
  • The distinction between use of the author’s own data on MCT1 expression consisting of 9 patients is mixed in with the much larger TCGA RNA expression metanalysis, and it is confusing about which data were which. Authors need the figures to more clearly differentiate which data are their own and which are derived from the TCGA cohort.
  • The selection and application of statistical assessments performed on the set of data obtained from the nine patients needs independent statistical verification.

Reviewer 2 Report

It is an interesting paper and well executed. Would be interested in knowing if you looked at serum LDH levels in the patients? Serum LDH levels have been shown previously to be associated with worse prognosis of patients with ccRCC (Shen, J. et. al. Plos One, 2016).   I would also like to know how the HP Lactate/HP Pyruvate value changes if you normalize to 1H ADC value or % perfusion fraction which also correlates with tumor grade. Can you add to your discussion how on how MCT1 can be expressed at the same level in both normal renal tissue (contralateral kidney) and is expressed at lower levels in some of your tumor samples compared to normal kidney tissue samples, but at the same time higher tumor grade and MCT1 expression is correlated? It seems to me that some other factor (possibly NADH levels) is contributing the differences seen in kPL and HP Lactate to HP Pyruvate ratios.  
